# A Mixed Visual Encoding Model Based on the Larger-Scale Receptive Field for Human Brain Activity

**DOI:** 10.3390/brainsci12121633

**Published:** 2022-11-29

**Authors:** Shuxiao Ma, Linyuan Wang, Panpan Chen, Ruoxi Qin, Libin Hou, Bin Yan

**Affiliations:** Henan Key Laboratory of Imaging and Intelligent Processing, PLA Strategic Support Force Information, Engineering University, Zhengzhou 450001, China

**Keywords:** visual encoding models, deep neural networks, receptive field, a large convolution kernel, RepLKNet, fMRI

## Abstract

Research on visual encoding models for functional magnetic resonance imaging derived from deep neural networks, especially CNN (e.g., VGG16), has been developed. However, CNNs typically use smaller kernel sizes (e.g., 3 × 3) for feature extraction in visual encoding models. Although the receptive field size of CNN can be enlarged by increasing the network depth or subsampling, it is limited by the small size of the convolution kernel, leading to an insufficient receptive field size. In biological research, the size of the neuronal population receptive field of high-level visual encoding regions is usually three to four times that of low-level visual encoding regions. Thus, CNNs with a larger receptive field size align with the biological findings. The RepLKNet model directly expands the convolution kernel size to obtain a larger-scale receptive field. Therefore, this paper proposes a mixed model to replace CNN for feature extraction in visual encoding models. The proposed model mixes RepLKNet and VGG so that the mixed model has a receptive field of different sizes to extract more feature information from the image. The experimental results indicate that the mixed model achieves better encoding performance in multiple regions of the visual cortex than the traditional convolutional model. Also, a larger-scale receptive field should be considered in building visual encoding models so that the convolution network can play a more significant role in visual representations.

## 1. Introduction

Understanding the brain’s information perception and information processing mechanisms when receiving stimuli is an important topic in traditional neuroscience research. Such information processing mechanisms are also crucial to artificial intelligence [1]. In neuroimaging research, researchers often use functional magnetic resonance imaging (fMRI) to construct a human visual encoding model [2,3]. fMRI uses a non-invasive method for research work that captures brain activities and encodes information about visual stimuli [4,5,6]. In such a visual encoding model, all non-linear characteristics are concentrated in the feature space, and a linear model can describe the relationship between the feature module and the subsequent voxel response [7]. The feature space model occupies an important position in the visual encoding model and is an essential factor in the final encoding performance.

From the perspective of modern neurological research, the population receptive field (PRF) in different visual areas of the brain is different, and the PRFs in low-level visual areas are generally fewer than those in high-level visual areas [8]. In 2007, Serge et al. proposed a method that can quantitatively calculate the PRF size of the visual cortex [8]. They found that the size increased four times between V1 (the primary visual cortex [9]) and V3 (the extrastriate cortex [9]). The size increased five to six times between V1 and LO (lateral occipital) [8]. It can be concluded that the PRFs of the higher visual cortex (HVC) of the brain are much larger than those of the lower visual cortex (LVC). In other words, the structure of the feature space in the encoding model with a large-scale receptive field is very suitable for actual biological discoveries.

Early research on visual neural encoding models mainly built the feature space with hand-made non-linear features. The article [10] published by Hubel et al. in 1962 suggested that the receptive field size of the low-level visual units of the cerebral cortex is small, illustrating the characteristic that the voxels in this area are very sensitive to low-level features. Moreover, the primary visual area was simulated by the Gabor wavelet model by setting different parameters such as position and orientation. Kendrick N. Kay et al. proposed a low-level visual area encoding model based on Gabor in 2008 [11]. Theoretically, the encoding model obtains better encoding performance when technical means are used to increase the corresponding RF area in the visual encoding model according to the trend from LVC to HVC.

Neural networks have been widely used by many researchers in different applications such as face recognition, medical applications, manufacturing, and economics [12]. The researchers from Oxford University proposed the VGG network model in ILSVRC (ImageNet Large Scale Visual Recognition Challenge) 2014 [13]. The VGG model proves that increasing the depth of the network can significantly improve the model’s performance. However, blindly increasing the network depth will bring problems such as gradient disappearance, gradient explosion, and performance degradation. To solve these problems, He et al. proposed the ResNet network model in 2015 [14], which uses batch normalization and shortcut connections. The addition of these two techniques made ResNet achieve first place in ILSVRC (ImageNet Large Scale Visual Recognition Challenge) 2015. The VGG and ResNet model illustrates that the application of CNN in computer vision is very successful. Based on this, more researchers are trying to apply CNN in visual encoding.

The visual encoding based on the features of the deep learning model mainly uses the feature data of the image extracted by the deep learning model as the medium to map between the natural image stimuli and the visual cortex voxel response. In 2014, Agrawal et al. [15] exploited a pre-trained CNN model to build a visual encoding model. The CNN model is pre-trained on ImageNet through the image classification task. In this approach, the pre-trained CNN model has strong image feature extraction ability, making the model perform well in visual encoding. Since this modeling approach usually separates image feature extraction and stimuli-to-voxel mapping, it is called a “two-step” model. Additionally, researchers are working on a unified optimization scheme from image feature extraction to the subsequent relationship mapping. In 2020, Qiao et al. [16] proposed an end-to-end model. Although the model only uses three convolutional layers and one fully connected layer, the end-to-end model performs better in low-level visual encoding. Since the model can encode the entire visual area at once instead of encoding for voxels, it achieves high encoding efficiency.

A key factor for the outstanding encoding performance of CNN models in visual encoding is that the CNN model can expand the effective receptive field (ERF) area by increasing the network depth and adopting other non-intuitive approaches [17,18,19,20,21]. Unlike the fully connected neural network model, where each unit value depends on the entire input, a unit in the convolutional network depends on only one region of the input. This area is the ERF of the unit. As a fundamental concept of CNN, ERFs are essential for understanding and diagnosing the working depth of CNNs. Since any location in the input image outside a unit’s ERF does not affect the value of the unit, it is necessary to carefully control the ERF to ensure that it covers the entire relevant image area [22]. However, the traditional deep learning model is inadequate to expand the receptive field by increasing the depth.

Ding et al. proposed a RepLKNet network model [23] that expands the ERF size by increasing the convolutional kernel size based on structural reparameterization. The new model enlarges the traditional convolution kernel size to that of a large-scale convolution kernel. The article by Ding et al. reveals that large-scale convolutional kernel networks are still effective in events. RepLKNet adopts methods such as structural reparameterization [24,25,26,27,28] to improve the performance of large convolution kernel networks for downstream tasks. The researchers demonstrated that directly using several large kernels instead of many small kernels can more effectively generate a larger effective receptive field and improve the performance of CNNs.

The previous description indicates that the visual encoding model can be realized with small convolution kernels, but its ERF area is small, which leads to its defects in acquiring image features; In modern biological research, the PRFs of HVC are much larger than those of the LVC; The RepLKNet model improves the ERF area by increasing the size of the convolution kernel, which is intuitive and efficient. Based on these observations, this paper formulates a hypothesis that when the receptive field of the convolutional layer in the encoding model has a larger size, the model can meet the physiological needs of a larger receptive field in HVC, and the model can capture more information, thus improving the performance of the model in visual encoding tasks. In the experiment, the encoding effect of the model with a large-size convolution kernel is better than that of the small-size convolution kernel model, which demonstrates that our hypothesis is true.

Therefore, this paper proposes a mixed model based on a large convolution kernel model and a traditional CNN model to optimize the feature space of the visual encoding model. By introducing the large convolution kernel model into the traditional CNN model, the mixed model expands the ERF size and obtains richer information than the CNN model.

The contribution of this paper is to introduce convolutional neural networks with larger convolution kernels into visual encoding tasks to improve the performance of visual encoding models. The coexistence pattern of ERFs brought about by the cooperation of large and small convolution kernels is more consistent with the fact that each visual area of the brain includes receptive fields of different sizes. Our model focuses on explaining, comparing, and analyzing the performance of the encoding model from the ERF perspective, which provides new perspectives (e.g., the convolution kernel size) for future research.

## 2. Methods

### 2.1. The Overview of the Mixed Visual Encoding Model

This paper uses a mixed model based on RepLKNet and VGG16 models to construct a visual encoding model. The specific methods are described as follows:

Initially, the visual encoding model uses a pre-trained large convolution kernel model and a pre-trained small convolution kernel model to extract image feature data from the training and test sets. The mixed model then mixes the feature data extracted by the two pre-trained models into one feature map. Subsequently, a linear regression model mapping image features to visual voxel responses is established for each visual region, and the linear regression model is trained with the training set. Next, the linear regression model is used to infer the voxel responses of the image features in the test set. Finally, the correlation between predicted voxels and actual voxels is calculated. By comparing the correlations, the encoding ability of different encoding models can be evaluated. The whole procedure of the mixed visual encoding model is shown in Figure 1. In Section 2.1 and Section 2.2, the feature space composed of mixed models and the linear mapping algorithm will be introduced in detail.

### 2.2. The Mixed Model

It can be seen from Figure 1 that the mixed model is the core part of the mixed encoding model to accomplish the image feature extraction task. This paper proposes a mixed model based on a larger convolution kernel and a smaller convolution kernel. The main reason for this design is that the larger convolution kernel has a wider receptive field, while the smaller convolution kernel has good performance in capturing texture features. Therefore, combining the advantages of the two types of convolution kernels is a natural approach. The process of creating a mixed model is shown below.

The image dimension of our input model is 3 × 224 × 224 (channel × width × height). First, the RepLKNet [23] model consists of one stem, four stages, and several transition blocks in the middle of each main stage. Among them, the stage involves a large-scale convolution kernel module, which is a crucial part of the RepLKNet model. Therefore, the outputs of the four stages are taken as a part of the feature output of RepLKNet. Meanwhile, it was found through experiments that adding a regularization layer after the output results can improve the model’s performance. So far, the outputs of four stages and the outcomes after regularization have been selected to form five-layer feature data. The dimensions of each layer of data are 125 × 56 × 56, 256 × 28V28, 512 × 14 × 14, 1024 × 7 × 7, and 1027 × 7 × 7. This type of data is called R features, which is the result of feature extraction by RepLKNet.

The VGG16 [13] model consists of 13 convolutional layers (hidden layers) and three FC layers. Among these convolutional layers, the combination of two 64-channel 3 × 3 convolutional layers is called Block A, the combination of two 128-channel 3 × 3 convolutional layers is called Block B, the combination of three 256-channel 3 × 3 convolutional layers is called Block C, the combination of three 512-channel 3 × 3 convolutional layers is called Block D, and the combination of three 512-channel 3 × 3 convolutional layers is called Block E. The 13 convolutional layers are divided into five blocks. The outputs of these five blocks and the output of Block E after passing through the maxpool layer are taken as the feature output of the VGG16 model. These six layers of features are called V features, and the dimensions of the V feature are 64 × 224 × 224, 128 × 112 × 112, 256 × 56 × 56, 512 × 28 × 28, 512 × 14 × 14, and 512 × 7 × 7, respectively.

A hybrid operation is performed on the V and R features. When features are mixed, a stacking or merging strategy can be adopted. Since the dimensions of the V and R features are not the same, the stacking strategy cannot be applied directly, and the merging method is the best choice in this case. When the V feature and the R feature are merged, the resulting new feature is called the M feature.

The whole procedure of the mixed model is shown in Figure 2.

### 2.3. Voxel-Wise Linear Regression Mapping

In Figure 1, linear mapping realizes the transformation from feature data to voxel responses, so it is an important block in the visual encoding model.

Linear regression models are built for each voxel according to the following mathematical formula:(1)V=Fw+b
where, V is a m×1 matrix representing the inferred voxel value, and m indicates the number of samples. F is a m×(n+1) matrix representing the image feature data of the mixed model, n describes the feature number, and b is a constant. w is a (n+1)×1 matrix obtained from model training, and it represents the linear model weight. The feature number n is generally larger than the number of samples m, indicating that the feature contains a certain noise. Consequently, the features are not all useful for encoding the model, and some processes need to be undertaken to regularize the feature. Initially, the image features are reduced into m−1 dimensions through Principal Component Analysis (PCA), and the default parameters of the PCA function in MATLAB are used. For the input of image features with a dimension of m×n, the data processed by PCA has a dimension of m×(m−1). In the m−1 principal components of each data, the proportion of the data analyzed by each principal component is not far from the difference. Therefore, the data with a dimension of m−1 are chosen, and sparse regularization will be applied to the feature data after PCA. Subsequently, the ROMP algorithm (Regularized Orthogonal Matching Pursuit) is adopted to fit the voxel responses and feature data in the training dataset to obtain a sparse linear regression model. After the model is trained, it can infer the voxel response from the test set.

## 3. Experimental and Results

### 3.1. Experimental Data

**Experimental design.** The public dataset published by Kamitani et al. [29] is used in this paper. The dataset involves two types of experiments: an image presentation experiment and an imagery experiment. This paper only uses the data generated from the image presentation experiment. In this experiment, the researchers collected densely sampled functional MRI (fMRI) data from five participants (one female aged 23–38). The images viewed by the subjects were from the ImageNet database. The image presentation experiment consists of training data and test data, and each involves 24 and 35 fMRI runs (9 min and 54 s per run), respectively. Each run contains 55 blocks, including 50 blocks with different images and five randomly distributed repeating blocks that present the same image as the previous block. There are 33-s and 6-s rest periods before and after each run. The image is placed at the center of the display, blinking at 2 Hz for 9 s. 

**Data Sets.** In the training set, 1200 images covering 150 categories (eight images per category) are shown. In the test set, 50 images covering 50 types (one image per category) are presented, and the test set images are presented 35 times. Differing from the training images, the test images are of different classes, and the order in which the stimuli are presented is random.

**Data preprocessing.** First, to ensure a stable magnetization state of the collected data, the first 8-s scans for experiments (retinotopy experiment) in each round were removed. Similarly, the first 9-s scans for other experiments were discarded. Head motion correction was then performed on the EPI imaging data using SPM5. Next, the functional image data were registered with the high-resolution structural image data. Finally, the registered data were interpolated by using 3 × 3 × 3 mm^3^ voxels, and the voxels within each run were normalized over time for experimental data from image presentations.

**ROI selection.** A retinotopy experiment was conducted to delineate the boundaries between each visual cortex to identify V1–V4 regions in the subjects. Next, a localizer experiment was conducted to divide the LOC, FFA, and PPA regions. The experiment defines the voxel set from V1 to V3 as LVC; the LOC, FFA, and PPA voxels are defined as HVC. LVC, V4, and HVC are collectively defined as VC (visual cortex).

A detailed introduction to this data can be found in reference [29].

### 3.2. Experimental Configuration

#### 3.2.1. The Pre-Trained Model

The RepLKNet model pre-trained on ImageNet is adopted, and specific parameter settings during training are presented in Table 1. In our experiments, nine graphics cards (Nvidia TITAN RTX) in a single node are used for pre-training operations.

The pre-trained model is available at https://pan.baidu.com/s/1gspbbfqooMtegt_DO1TUeA?pwd=lknt, accessed on 27 November 2022.

Ross Wightman from Canada established the timm library on GitHub to help researchers to obtain standard neural network models. The information about the timm library is available at https://github.com/rwightman/pytorch-image-models, accessed on 27 November 2022. The VGG16 pre-trained model is also included in timm as a common CNN model. Therefore, the ‘’create_model’’ function in the timm library is used, and the pre-trained parameter in the function is set to true to obtain the pre-trained model. The pre-trained VGG16 model can be downloaded from https://download.pytorch.org/models/vgg16-397923af.pth, accessed on 27 November 2022.

#### 3.2.2. Comparison Models

To more comprehensively compare the performance difference of the feature space in the encoding model, several convolutional models and the RepLKNet model are taken for comparison.

**Comparison Methods Group 1: Baselines.** Group 1 is a traditional visual encoding model. Traditional convolutional models, mainly VGG16 and ResNet50, are used as the image feature extractor. Subsequent operations include training a linear regression model, predicting voxel values, and calculating the correlation between predicted and true voxel values. 

From these descriptions, it can be seen that this baseline model only differs from our model in feature extraction, and the other steps remain the same.

**Comparison Methods Group 2: RepLKNet.** With the RepLKNet model proposed by Ding et al. [25], the large-scale convolution kernel model has received the attention of researchers again. It has been demonstrated that directly using larger convolution kernels instead of stacking small convolution kernels can obtain larger ERFs, thereby improving CNN performance.

To compare with Group 1, RepLKNet is used as a feature extractor for visual encoding models in our experiments, and other operations are the same as those in Group1.

**Comparison Methods Group 3: End-to-End Model.** Although the above two models use different network models when extracting features, their essence is the same. The end-to-end model is a brand-new visual encoding model.

By integrating the image representation model and the voxel regression model, the end-to-end training method proposes an end-to-end convolutional regression network-based visual encoding model (ETE-CRNVEM). ETE-CRNVEM includes two parts, where the convolutional layer (conv) in front of the network is used for image representation (i.e., the S2F module), and the final fully connected layer (FC) is used for voxel regression (i.e., the F2V module). In the training process of the end-to-end model, the feature extraction process is no longer separated from the voxel mapping process, but the parameters of the two-in-one model are unified for optimization. Consequently, the end-to-end model can learn the complete process from image stimuli to voxel regression. In this way, the image representation module and the voxel regression module can cooperate during the optimization process to extract image features that better match the voxel responses and achieve better encoding performance. Please refer to references [16,30] for more details.

### 3.3. Evaluation Strategy

For the visual encoding model, the prediction correlation is adopted to evaluate the model performance, and the accuracy of the method is calculated as follows:(2)$$Pcc=cor\left(v,\hat{v}\right)$$
where, Pcc represents the Pearson correlation coefficient (PCC) [31,32], and its value falls between the actual voxel v and the forecasted voxel $\hat{v}$ for all image features in the test set. There is a positive correlation between the Pcc value and the model encoding performance. In this paper, voxels with correlation values greater than 0.41 are defined as valid predicted voxels [8].

### 3.4. PCC Results for Different Models

In this paper, feature collection and voxel mapping operations are performed on five subjects for ResNet50, VGG16, end-to-end models, and integrated models, respectively, and the performances of those models are finally obtained.

Table 2 shows the top 100 voxel correlation values of subject 3. It demonstrates that the mixed model achieves better performance than the VGG16, ResNet50, and End-to-End models. For example, in the V1 region, compared with VGG16 and ResNet50, our model significantly improved the performance in predicting voxel correlations.

Figure 3a,b show the comparison results of the encoding performance of each brain region between different models for subject 3 in the fMRI dataset. In Figure 3, the encoding performance of VGG16, ResNet50, and the mixed model is compared through a scatter plot. The scatter plot directly selects the predicted correlation value of a single voxel as the drawing target so that the difference between different models can be observed in the figure. For example, in the V1 map of Figure 3a, it can be seen that the red points are significantly greater than the blue points, indicating that the mixed model has better encoding performance.

Similarly, the encoding results of different models in the brain regions of subject 3 are listed in Table 1. It can be seen from this table that the mixed model has obvious advantages in all brain regions, and our model has an average of 20% encoding performance improvement in different brain regions compared to the traditional model. This result is also confirmed in Figure 3a,b. In the scatter plot, the red points are about 10–30% more than the blue points. From the above table and scatter plot, it can be concluded that the mixed model has a better encoding effect than the traditional convolutional neural network.

## 4. Discussion

### 4.1. The ERF and the Convolutional Kernel Size

Based on the experimental data, it can be found that the mixed model that mixes RepLKNet and VGG16 achieves better encoding performance than ordinary VGG16 or ResNet. The possible reasons are discussed and analyzed as follows.

The ERF size of traditional CNN networks can be increased in several ways. One option is to stack more layers and make the network deeper, thus theoretically increasing the size of the receptive field linearly. This is because with each additional layer, the receptive field size increases with the convolution kernel size. Meanwhile, subsampling increases the receptive field size multiplicatively. The VGG model [13] and ResNet [14] combine the above techniques to enlarge the ERF size. However, the ERF obtained in this way is not as large as that obtained by directly expanding the convolution kernel size in the RepLKNet model.

To show the difference between traditional CNN and RepLKNet in ERF more intuitively, quantitative methods are exploited to calculate the ERF size. This approach calculates the contribution score of the input image’s feature map to the last layer of the convolutional model. These scores are then displayed as a heatmap, which can be shown as the ERF size of the CNN in Figure 4. See [22] for details. Based on this algorithm, an image that intuitively shows the difference in ERF between different models is obtained.

It can be seen from Figure 4A–C that although the depth of the ResNet model gradually increases from 18 layers to 101 layers, the ERF range shown in the green part of the figure does not increase significantly. The same result can be observed in the VGG16 and VGG19 models, as shown in Figure 4D,E. However, in Figure 4F, the receptive field of RepLKNet is vast compared to VGG and ResNet, indicating that the ERF of the RepLKNet model is more significant than that of the traditional CNN model.

From Figure 5, it can be concluded that the ERF size of visual encoding models using traditional convolutional neural networks is insignificant. To address this issue, a large convolution kernel model represented by RepLKNet is introduced into the traditional convolution model, and the mixed model is proposed. The mixed model enables traditional models to obtain a “huge” receptive field range to improve the ability to obtain feature information and ultimately improve encoding performance.

Table 1 indicates that for the original VGG16 and ResNet50 models, the top 100 Pcc values of the two models for the seven brain regions are basically the same. For our mixed model, the top 100 correlation values for the seven brain regions are improved to varying degrees.

Meanwhile, it can be seen that in visual encoding, introducing a larger ERF into traditional CNN models is a crucial process, which is in line with biological research findings. This method can ensure that the model has texture bias due to the traditional convolution kernel and obtains a larger receptive field to ensure the acquisition of more information (such as shape bias).

Thus, introducing a larger ERF into traditional CNN models guarantees that the mixed model achieves better encoding performance in experiments.

### 4.2. V1 Area Performance and Advanced Area Performance

As mentioned earlier, high-level visual areas have larger receptive fields than LVC. Therefore, theoretically, the encoding effect of the mixed model in the HVC should be better than that in the LVC, but this is inconsistent with the previous theory according to the experimental results. This phenomenon is discussed below.

Generally, there are two processing procedures within 100 ms [33] when subjects perform picture stimulation. These two information procedures are often referred to as “top-down processing” and “bottom-up processing” [34]. When the researchers collected data on the subjects, the MRI equipment’s repetition time (TR) was set to 3 s, indicating that the subjects’ visual areas underwent 30 such information processes in one cycle. Specifically, the fMRI response data includes data from several areas of the visual cortex. According to neuroanatomy and physiology studies [29,34], the primary visual cortex receives and processes information from the visual information relay station LGN, then transmits the information to V2 and V3 for processing, and then transmits the processed information to V4 and higher-level visual areas. V2 receives the feedforward information from V1 and passes it to higher-level brain areas such as V3, V4, etc., and it also has feedback connections to V1. V3 is in front of V2, and it receives information from V1 and V2 and projects the information to the posterior parietal cortex. V4 is located in front of V2 and receives the information from V2 and V3. V4 also receives input from V1, especially the central part. In the ventral visual pathway, the visual information flows via V1 to V2 or V4 and then to higher-level visual areas, such as the LO. The specific connection relationship is presented in Figure 5.

It can be seen from Figure 5 that the V1, V2, and V3 regions in the LVC can all obtain information from the HVC through neural pathways. This is one of the reasons why the encoding performance of the LVC is better.

### 4.3. Limitations and Future Work

Although our mixed model can improve the visual encoding performance of each visual area, it still has shortcomings. In the design of the mixed strategy, RepLKNet and VGG16 are combined through the merge method, and the weights of these models are the same. This paper fails to discuss the difference caused by different weight values. In future research, our work will focus on optimizing the weights of the two models and designing a more efficient model mixing strategy.

## 5. Conclusions

This paper proposes a mixed model that mixes the RepLKNet with VGG16. The proposed model improves the receptive field size of the CNN model used for feature extraction in previous visual encoding models. Meanwhile, it ensures good performance in the brain’s visual areas by accumulating information from the large receptive field while retaining the performance advantage of traditional convolutional networks. The experimental result confirms that expanding the receptive field of the convolution kernel can effectively improve the encoding performance of the model. The mixed model focuses on improving the encoding model’s performance from the ERF perspective, which provides a new approach for future visual encoding studies.

## Figures and Tables

**Figure 1 brainsci-12-01633-f001:**
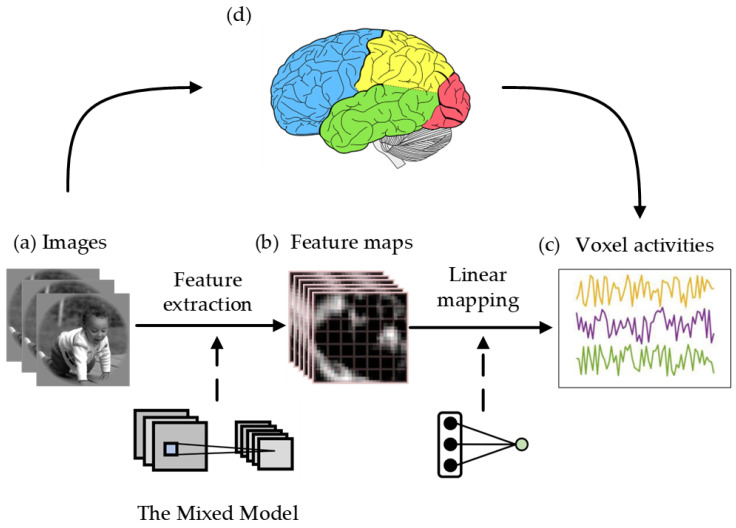
The mixed visual encoding model. (**a**) Images; (**b**) The feature maps from the pre-trained mixed model; (**c**) The voxel activities; (**d**) Information processing of the visual cortex in the brain. When the brain’s visual cortex processes external images, fMRI can measure activity responses. Meanwhile, the linearized visual encoding methods extract image features from pre-trained mixed models. The visual encoding model uses the fMRI and image feature data in the training set as input to train the linear regression model. Then, the linear model obtains the inferred fMRI voxel by inputting the image feature map from the test set. Finally, the visual encoding model calculates the correlation.

**Figure 2 brainsci-12-01633-f002:**
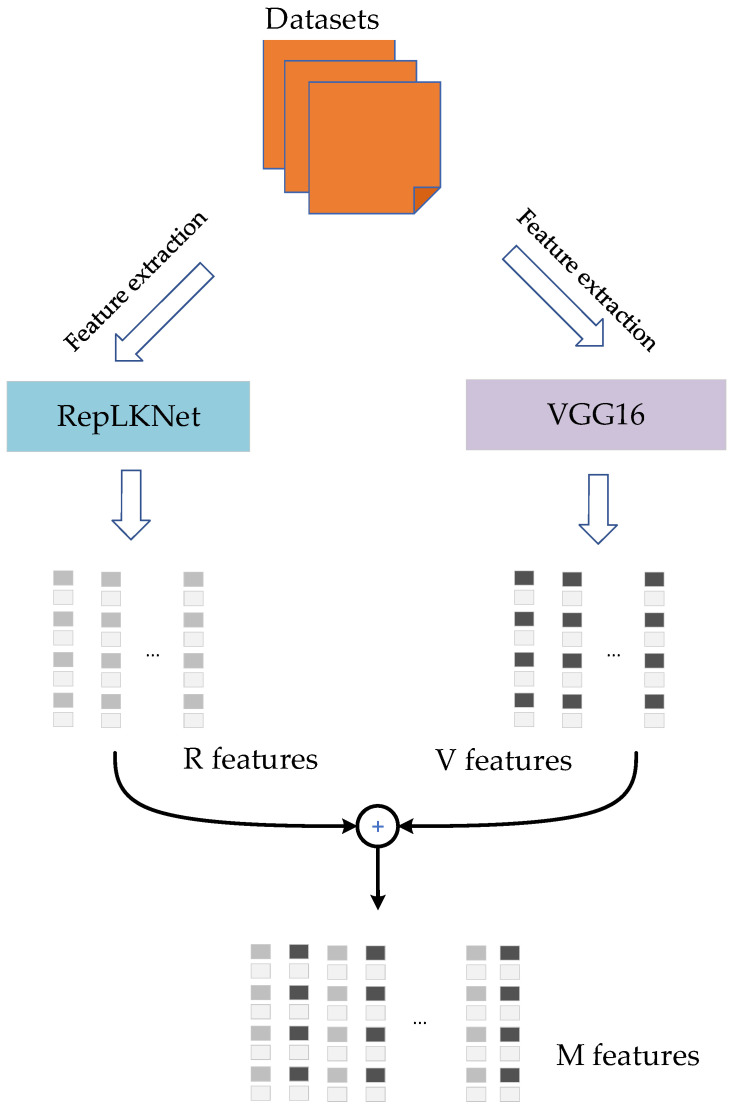
The mixed model. The RepLKNet model is adopted to extract the feature data of the images in the dataset, which are called R features. Similarly, the image data extracted by the VGG16 model are called V features. The R feature and the V feature are then mixed to obtain mixed feature data with different receptive field sizes, called M features.

**Figure 3 brainsci-12-01633-f003:**
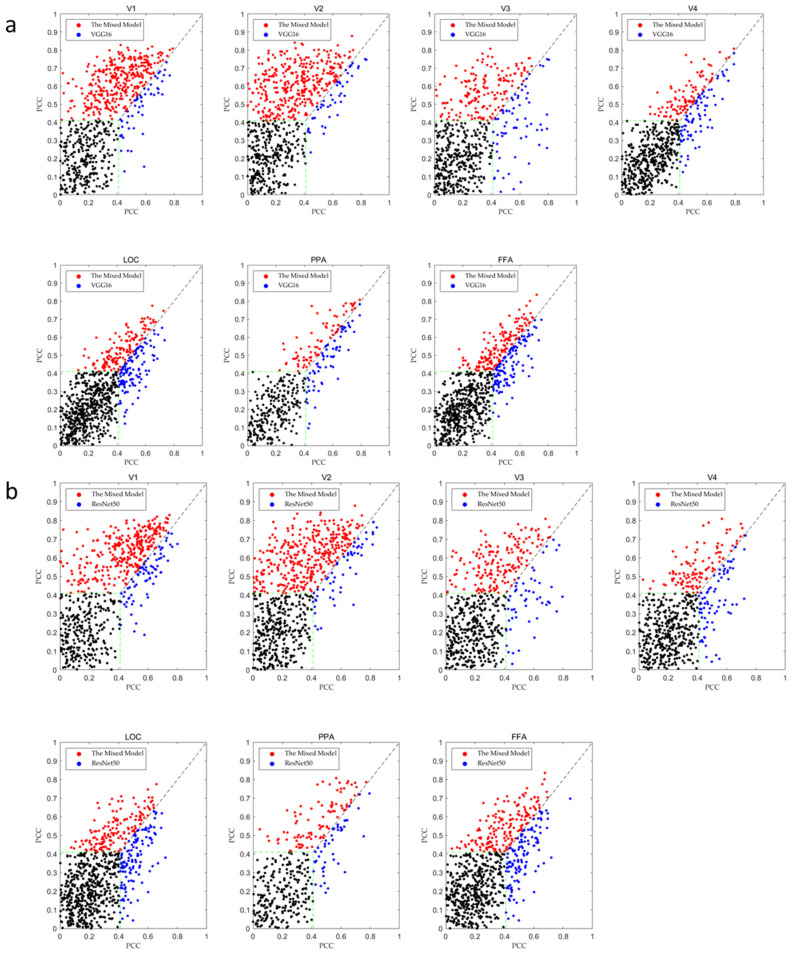
(**a**) Comparison of the encoding performance between the mixed model and VGG16. (**b**) Comparison of the encoding performance between the mixed model and ResNet50. The horizontal and vertical axes in the scatterplot represent the correlation values of the voxels predicted by the traditional convolution model and the correlation values of the voxels predicted by the mixed model, respectively. The red points represent the points at which the correlation of the voxels predicted by the mixed model is higher than the traditional model. The blue points represent the points at which the correlation of the voxels predicted by the traditional model is higher than that of the hybrid model. The black points represent the points at which the voxel correlation predicted by the two models cannot reach a valid value. The green line in the scatterplot represents the threshold for valid predicted voxels (i.e., 0.41).

**Figure 4 brainsci-12-01633-f004:**
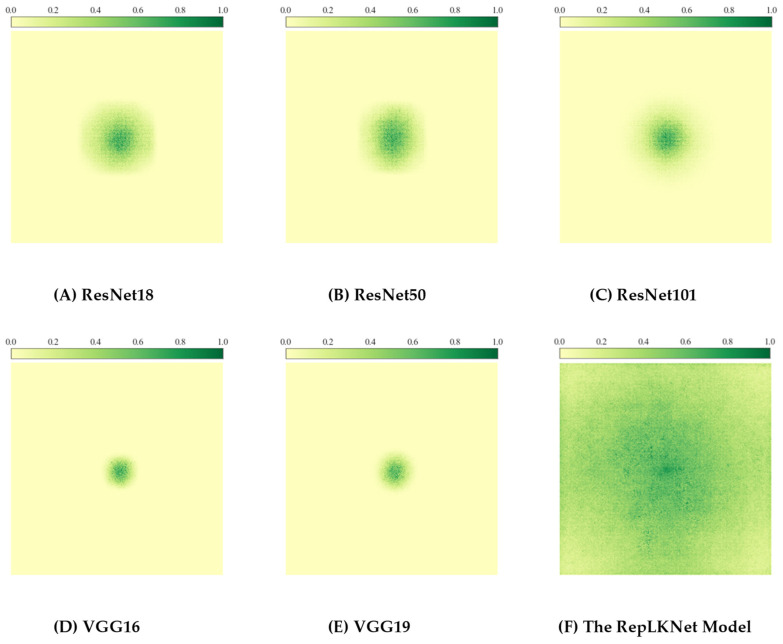
The difference in ERF between different models.

**Figure 5 brainsci-12-01633-f005:**
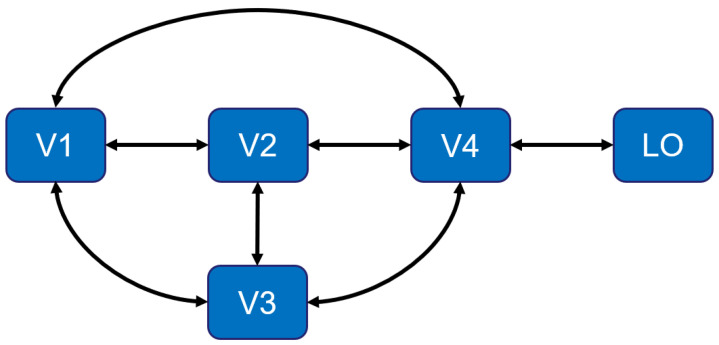
The diagram of connections between brain regions.

**Table 1 brainsci-12-01633-t001:** The parameters of the RepLKNet model during the training phase.

Parameters	Value
Batch size	32
Drop path	0.1
LR	4 × 10^−3^
Warmup epoch	5
Epoch Number	90

**Table 2 brainsci-12-01633-t002:** The ROI-level encoding performance of VGG16, ResNet50, End-to-End, and our models of subject 3. Bold fonts mark our data results.

Areas	Models				
	VGG16	ResNet50	End-to-End	RepLKNet	Ours (Mixed Model)
V1	0.620	0.650	0.683	0.741	**0.743**
V2	0.617	0.645	0.644	0.740	**0.746**
V3	0.576	0.557	0.559	0.647	**0.650**
V4	0.569	0.547	0.300	0.521	**0.582**
LOC	0.573	0.566	0.311	0.490	**0.589**
PPA	0.590	0.546	0.233	0.545	**0.601**
FFA	0.601	0.589	0.320	0.512	**0.624**

## Data Availability

The fMRI data set involved in this paper is available at the following website: https://github.com/KamitaniLab/GenericObjectDecoding (20 July 2021).

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
