# Peer review of "A Mixed Visual Encoding Model Based on the Larger-Scale Receptive Field for Human Brain Activity"

_brainsci, 2022, doi:10.3390/brainsci12121633_

Round 1

Reviewer 1 Report

This is an article about an interesting subject, which is the implementation of a mixed visual encoding model based on the larger-scale receptive field for human brain activity. The manuscript is generally well structured.

English language and style are fine but there are some minor issues that need to be addressed before publication (e.g. in line 334 the term “research researchers” needs rephrasing). 

The introduction describes the background of this study in a comprehensive manner.

“Materials and methods” section is descriptive enough and well-structured too.

The results are very interesting and, to my opinion, well presented.

I think that a paragraph summarizing the limitations of this study should be added in the discussion section to further improve the quality of this manuscript.

Perhaps conclusions should be written in a more detailed manner, making the key points clearer and more comprehensible to the readers.

Author Response

Dear Reviewer,

Our response to your suggestion is provided in the attachment for your review.

Thank you and best regards.

Yours sincerely, 

Shuxiao Ma

Reviewer 2 Report

General comment: 

This work deals with the use of CNN in fMRI for increasing the effective receptive field of visual neural coding of multiple visual cortex regions. 

Specific comments throughout the paper:

1. Introduction 

Lines 29-32: missing ref.

Line 45: Remove the "*" if there is no footnote associated to this symbol. 

Lines 46-47: Please define V1, V3, LO, consider that the readership may not be completely specialized in this field. Hence, if you open the writing to a wider audience, the visibility and value for the journal. 

Lines 53-58: Missing ref. Probably the authors forgot to refer to https://doi.org/10.1038/nature06713

Lines 59-63: Good point. 

The writing is quite fluent and the knowledge gap is well stressed. Anyway, the referencing can be improved. 

2. Methods

Fig. 1 is very descriptive. Thank you for providing it. 

Line 119: Why there isn't a ref. for Vgg16?

In Eq. (1) vector or matrix terms are not clearly and explicitely defined in terms of notation (e.g., bold or underlined). Please clarify and be coherent with the following text. 

Lines 156-157: More details about the PCA compression has to be provided. 

3. Experiments and results

The information in sect. 3.1 are more appropriate to the methods. I suggest the authors to reconsider the manuscript organization

Also, Sect. 3.2 is a very methodological one. I suggest to split and divide the results for a better presentation of them. 

The correlations value reported in Tab. 1 are relatively low. The authors are not comparing them to other values from the literature. Please support the validity of your findings. 

Fig. 4: I suggest the authors to provide a legend for the colors. Also, improve the figure quality by providing the labels for x and y.

4. Discussion

A discussion section is provided, which is a pros.  

Missing caption for Fig. 5. The plots are not clearly interpretable. Please improve this part. 

5. Conclusions

The conclusion section can be improved by providing more future perspectives.

Minor edits:

Line 119: Missing space.

Line 201: No indent after the equation.

Author Response

(The authors gave the same response as above.)

Reviewer 3 Report

In the paper titled “A Mixed Visual Encoding Model Based on the Larger-scale Re-ceptive Field for Human Brain Activity”  the authors paper proposes a linearized visual coding method to construct a feature extraction  model based on a mixture of large convolution kernels and small convolution kernels.

Abstract: The readers are confused and unable to comprehend the purpose of this manuscript due to poor description and a lack of information about the topic.

Introduction: Refocusing and a more thorough definition of the argument presented in this paper, in my opinion, are both necessary. Readers become perplexed about the goal when the issue is not well described. Additionally, I advise hiring a native English speaker to edit the paper.

Intrduction: “In March 2022, Ding et al. proposed a RepLKNet network [18]”. I would advise trimming and refocusing on "In March 2022."

In line 79 I would recommend clarifying “HVC ”

I would recommend a related work section

In “In 2014, the University of Oxford and Google proposed a deep convolutional neural net- 138 work: VGG. In 2015, He Kaiming's team proposed the Resnet model.” I would recommend refocusing and referencing.

Sentence in line 158-159 is not clear. Which linear regression models are most frequently employed?

In line 166 I would recommend reference

I would recommend adding more information to better characterize the data used in this research paper as well for dataset organization.

I would suggest improving the data preprocessing part.

I would suggest improving dataset preparation, training, test and validation framework used in this paper

 I would suggest using ROC curve

I would suggest add info on hyperparameters, architecture information, number of epochs, learning rate, etc. 

I advise utilizing a metric to report the effectiveness of the models employed in this study.

Author Response

(The authors gave the same response as above.)

Reviewer 4 Report

Paper: A Mixed Visual Encoding Model Based on the Larger-scale Receptive Field for Human Brain Activity

This paper proposes a mixed model, which mixes the RepLKNet with Vgg16 for human brain activity. The main point is that the paper is not clear in some points. Therefore, I ask the authors to discuss every section in detail and clearer. In addition, the following comments can help the authors to improve their paper and to be in a good form.

Comments:

P1: The paper suffers from long sentences which make confusion. Improve the writing of the paper.

P2: The abstract is superficial. Deep information about the proposed methodology should be included.

P3: The literature review presented in the introduction part is short. Authors should discuss it in detail and use more references. In addition, the main contribution of the paper should be improved and presented clearer.

P4: It is very important to add figures presenting the used data whether for training or validation in (subsection 3.1. Datasets). Some analysis for these data can be also added and presented.

P5: Are the used data for validation enough?

P6: More information about the training and validation processes should be added.

P7: Comparisons with other related works should be added in (section 4. Discussion).

P8: All sections of the paper require a detailed discussion. Try to present everything clearly and in detail. Flowchart also can be added to show the executed code.

P9: The following paper is very related to the topic of the paper. The authors can read it carefully and use it in their paper: https://doi.org/10.15377/2409-5761.2020.07.2

P10: The conclusion is not written well. Rewrite it and add some future work.

P11: Revise the English of the paper carefully. There are many errors in the paper.

Author Response

(The authors gave the same response as above.)

Round 2

Reviewer 2 Report

I thank you for your time and patience in answering to my questions and comments, while thoroughly reviewing your work and improving it. 

Few issues has to be solved:

About the PCA compression, with my comment, I wanted to know which model parameters were used and more technical details, not a general explanation of the algorithm. 

The colorbar in Fig. 5F is different in its min.-max. values. Please use the same colorbar. 

Author Response

(The authors gave the same response as above.)

Reviewer 3 Report

A Mixed Visual Encoding Model Based on the Larger-scale Receptive Field for Human Brain Activity" deals with a subject that is very interesting, but as I read through the sections, I began to have some concerns because the research topic hasn't been sufficiently covered. I want to emphasize

1 Language, writing, and spelling issues;

2 Flaws in the procedures, presentation, and analysis of the data;

3 Finally, neither a clear hypothesis nor sufficient results have been provided.

Author Response

Dear Reviewer,

Our response to your suggestion is provided in the attachment for your review. The revised manuscript has been grammatically revised by professionals, and we will also send you the revision certificate as an attachment.

Thank you and best regards.

Yours sincerely, 

Shuxiao Ma

Reviewer 4 Report

Please reconsider again my previous comments to improve the quality of the paper. In addition, It is very important to add figures showing the used or the experimental data whether for training or validation in (subsection 3.1. Datasets).

Author Response

(The authors gave the same response as above.)
